# Numerical Solution for Singular Boundary Value Problems Using a Pair of Hybrid Nyström Techniques

**Mufutau Ajani Rufai** [1,*,†] and **Higinio Ramos** [2,†,‡]

1   Dipartimento di Matematica, Università degli Studi di Bari Aldo Moro, 70125 Bari, Italy
2   Scientific Computing Group, Universidad de Salamanca, Plaza de la Merced, 37008 Salamanca, Spain; higra@usal.es
*   Correspondence: mufutau.rufai@uniba.it
†   These authors contributed equally to this work.
‡   Current address: Escuela Politécnica Superior de Zamora, Campus Viriato, 49022 Zamora, Spain.

**Abstract:** This manuscript presents an efficient pair of hybrid Nyström techniques to solve second-order Lane–Emden singular boundary value problems directly. One of the proposed strategies uses three off-step points. The obtained formulas are paired with an appropriate set of formulas implemented for the first step to avoid singularity at the left end of the integration interval. The fundamental properties of the proposed scheme are analyzed. Some test problems, including chemical kinetics and physical model problems, are solved numerically to determine the efficiency and validity of the proposed approach.

**Keywords:** hybrid block methods; chemical kinetics and physical models; Lane–Emden singular boundary value problems; starting procedure; convergence analysis



## 1. Introduction

In this paper, we consider the following two-point singular boundary value problem (SBVP):

$$y''(x) + \frac{\lambda}{x} y'(x) = k(x, y), \quad 0 < x \le x_N = 1. \tag{1}$$

We consider this together with any of the two-point boundary conditions:

$$y(0) = y_a, \quad y(1) = y_b, \tag{2}$$

or

$$y(0) = y_a, \quad y'(1) = y'_b, \tag{3}$$

or

$$y'(0) = y'_a, \quad y(1) = y_b, \tag{4}$$

where $\lambda, y_a, y_b, y'_a, y'_b$, are known real values and $k(x, y)$ denotes a continuous real function, where we assume the necessary conditions to guarantee the existence of a unique solution to the problem. The existence and uniqueness of the solution to the problem (1) subjected to any of the boundary conditions above have been established by Pandey [1] and Zou [2].

According to Thula and Roul [3], the mathematical expression of numerous problems arising in chemical kinetics, astrophysics, catalytic diffusion reactions, celestial mechanics, engineering, and various physical models gives rise to second-order singular boundary value ordinary differential equations (SSBODEs) of the type given in (1).

The problem of reactant concentration in a chemical reactor, reaction–diffusion processes inside a porous catalyst, the conduction of heat in the human head, the distribution of oxygen in a spherical shell, and many others, can be modelled by the system (1).

Significantly, much work has been carried out to obtain numerical solutions for the above singular problems. Different strategies have been reported for solving (1),

where the fundamental difficulty arises due to the singularity at $x = 0$. Notable scholars in the field of numerical analysis have proposed numerical techniques for solving the problem (1). Examples of such techniques include the finite difference methods (FDM) proposed in [4,5], the spline methods (SM) proposed in [6,7], the Padè approximation method (PAM) introduced in [8,9], the pseudospectral method (PM) proposed in [10], or the Jacobi–Gauss collocation method (JCM) reported in [11]. Other manuscripts on recently developed numerical or analytical techniques for solving (1)–(4) inlcude those of [12–18].

We propose a pair of hybrid Nyström techniques (PHNT) for solving the SBVPs given in (1) numerically. The main formulas provide a method of order six, which cannot solve the problem on its own due to the singularity at $x = 0$. We have designed a second ad hoc method that applies only to the first subinterval and is unaffected by singularity. In this way, we obtain a scheme capable of solving the problem posed effectively. Comparisons show that the proposed method is more advantageous than other existing methods. A problem of particular interest is how to select the optimal value of $h$. We have not addressed this issue here, but the CESTAC method and the CADNA library could be helpful for this task [19,20]. This technique is based on the use of stochastic arithmetic in place of floating-point arithmetic to validate the results and find an optimal solution.

The present work is outlined as follows. In Section 2, we present the PHNT method for solving SBVPs. The characteristics of the developed formulas are analyzed in Section 3. Some issues with the implementation of the PHNT are considered in Section 4. In Section 5, we present the numerical results of some physical models and catalytic diffusion–reaction problems to show the efficiency and reliability of the proposed technique. Some conclusions are outlined in Section 6.

## 2. Development of the PHNT Method

To obtain the PHNT method, we firstly reformulate the equation in (1) as $y''(x) = f(x, y(x), y'(x))$, where $f(x, y(x), y'(x)) = k(x, y(x)) - \frac{\lambda}{x} y'(x)$. Thus, the singularity is transferred to the function $f$.

### 2.1. Main Formulas

We consider that the exact solution $y(x)$ of the SBVP on an interval $[x_n, x_{n+1}]$, $x_n > 0$ with step size $h = x_{n+1} - x_n$ can be approximated by a polynomial $p(x)$ in the form:

$$y(x) \simeq p(x) = \sum_{j=0}^{6} a_j \, x^j, \tag{5}$$

From this, it readily follows that:

$$y'(x) \simeq p'(x) = \sum_{j=1}^{6} a_j j x^{j-1}, \tag{6}$$

$$y''(x) \simeq p''(x) = \sum_{j=2}^{6} a_j j(j-1) x^{j-2}, \tag{7}$$

where $a_j \in \mathbb{R}$ are unknown coefficients that must be specified using collocation conditions at some chosen nodes.

We take the following intermediate nodes on $[x_n, x_{n+1}]$: $x_{n+u} = x_n + uh$, $x_{n+v} = x_n + vh$ and $x_{n+w} = x_n + wh$ with $0 < u < v < w < 1$. Consider the approximations in (5) and (6) evaluated at $x_n$, as well as the one in (7) evaluated at $x_n, x_{n+u}, x_{n+v}, x_{n+w}, x_{n+1}$. Doing this, we obtain a system of equations with seven unknowns $a_n$, $n = 0(1)6$, given by:

$$p(x_n) = y_n \, , p'(x_n) = y'_n \, , p''(x_n) = f_n,$$
$$p''(x_{n+u}) = f_{n+u} \, , p''(x_{n+v}) = f_{n+v} \, , p''(x_{n+w}) = f_{n+w} \, , p''(x_{n+1}) = f_{n+1},$$

where $y_{n+j}$ and $f_{n+j}$ denote approximations of $y(x_{n+j})$ and $y''(x_{n+j})$, respectively. The system may be written in matrix form as:

$$
\begin{pmatrix}
1 & x_n & x_n^2 & x_n^3 & x_n^4 & x_n^5 & x_n^6 \\
0 & 1 & 2x_n & 3x_n^2 & 4x_n^3 & 5x_n^4 & 6x_n^5 \\
0 & 0 & 2 & 6x_n & 12x_n^2 & 20x_n^3 & 30x_n^4 \\
0 & 0 & 2 & 6x_{n+u} & 12x_{n+u}^2 & 20x_{n+u}^3 & 30x_{n+u}^4 \\
0 & 0 & 2 & 6x_{n+v} & 12x_{n+v}^2 & 20x_{n+v}^3 & 30x_{n+v}^4 \\
0 & 0 & 2 & 6x_{n+w} & 12x_{n+w}^2 & 20x_{n+w}^3 & 30x_{n+w}^4 \\
0 & 0 & 2 & 6x_{n+1} & 12x_{n+1}^2 & 20x_{n+1}^3 & 30x_{n+1}^4
\end{pmatrix}
\begin{pmatrix}
a_0 \\ a_1 \\ a_2 \\ a_3 \\ a_4 \\ a_5 \\ a_6
\end{pmatrix}
=
\begin{pmatrix}
y_n \\ y_n' \\ f_n \\ f_{n+u} \\ f_{n+v} \\ f_{n+w} \\ f_{n+1}
\end{pmatrix}.
$$

Solving this system, we obtain the values of $a_n$, $n = 0(1)6$. Using the substitution $x = x_n + zh$, the polynomial in (5) may be expressed as:

$$
\begin{aligned}
p(x_n + zh) \;=\; & \alpha_0(z)y_n + h\alpha_1(z)y_n' \\
& + h^2(\beta_0(z)f_n + \beta_u(z)f_{n+u} + \beta_v(z)f_{n+v} + \beta_w(z)f_{n+w} + \beta_1(z)f_{n+1}),
\end{aligned}
\tag{8}
$$

where $u = \frac{1}{3}, v = \frac{1}{2}, w = \frac{2}{3}$, and the coefficients $\alpha_0(z) = 1$, $\alpha_1(z)$, and $\{\beta_i(z)\}_{i=0,u,v,w,1}$ depend on $z$.

Evaluating the formula in (8) and its derivative at $z = 1$, we obtain approximations of $y(x_{n+1})$ and $y'(x_{n+1})$, given, respectively, by:

$$
y_{n+1} = y_n + hy_n' + h^2\left(\frac{9f_{n+u}}{20} - \frac{4f_{n+v}}{15} + \frac{9f_{n+w}}{40} + \frac{11f_n}{120}\right),
$$

$$
y_{n+1}' = y_n' + \frac{h}{120}(11f_n + 81f_{n+u} + 81f_{n+v} - 64f_{n+w} + 11f_{n+1}).
\tag{9}
$$

Now, evaluating $p(x)$ and $p'(x)$ at $x_{n+u}, x_{n+v}, x_{n+w}$, we obtain the following hybrid Nyström-type formulas:

$$
\begin{aligned}
y_{n+u} \;=\;& y_n - \frac{1}{3}hy_n' + \frac{h^2(240f_{n+u} - 224f_{n+v} + 87f_{n+v} + 83f_n - 6f_{n+1})}{3240}, \\
y_{n+v} \;=\;& y_n + \frac{h}{2}y_n' + h^2\left(\frac{207f_{n+u}}{1280} - \frac{f_{n+v}}{8} + \frac{63f_{n+w}}{1280} + \frac{163f_n}{3840} - \frac{13f_{n+1}}{3840}\right), \\
y_{n+w} \;=\;& y_n + \frac{2h}{3}y_n' + \frac{2h^2}{405}(51f_{n+u} - 32f_{n+v} + 15f_{n+w} + 12f_n - f_{n+1}).
\end{aligned}
\tag{10}
$$

$$
\begin{aligned}
y_{n+u}' \;=\;& y_n' + h\left(\frac{19f_{n+u}}{40} - \frac{152f_{n+v}}{405} + \frac{17f_{n+w}}{120} + \frac{329f_n}{3240} - \frac{31f_{n+1}}{3240}\right), \\
y_{n+v}' \;=\;& y_n' + \frac{h(1053f_{n+u} - 512f_{n+v} + 243f_{n+w} + 193f_n - 17f_{n+1})}{1920}, \\
y_{n+w}' \;=\;& y_n' + \frac{1}{405}h(216f_{n+u} - 64f_{n+v} + 81f_{n+w} + 41f_n - 4f_{n+1}).
\end{aligned}
\tag{11}
$$

The formulas in (9)–(11) altogether form the main block method.

### 2.2. Formulas to Circumvent the Singularity

This block method cannot be used directly for solving a BVP problem with the differential equation in (1) because it is not possible to evaluate $f_0 = f(x_0, y_0, y_0')$, since there is a singularity at $x_0 = 0$.

To overcome this drawback, we have developed a set of formulas specially designed for the subinterval $[x_0, x_1]$, where the value $f_0$ is absent. These formulas are obtained similarly to before as:

$$
\begin{aligned}
y_1 &= y_0 + hy_0' + h^2\left(\frac{51f_u}{40} - \frac{26f_v}{15} + \frac{21f_w}{20} - \frac{1}{120}11f_1\right), \\
y_1' &= y_0' + h\left(\frac{13f_u}{9} - \frac{16f_v}{9} + \frac{10f_w}{9} - \frac{f_1}{9}\right).
\end{aligned}
\tag{12}
$$

For the remaining formulas, we obtain:

$$
\begin{aligned}
y_u &= y_0 + \frac{h}{3}y_0' + h^2\left(\frac{329f_u}{1080} - \frac{194f_v}{405} + \frac{139f_w}{540} - \frac{89f_1}{3240}\right), \\
y_v &= y_0 + \frac{h}{2}y_0' + h^2\left(\frac{87f_u}{160} - \frac{193f_v}{240} + \frac{69f_w}{160} - \frac{1}{240}11f_1\right), \\
y_w &= y_0 + \frac{2h}{3}y_0' - \frac{1}{405}2h^2(-159f_u + 224f_v - 123f_w + 13f_1).
\end{aligned}
\tag{13}
$$

$$
\begin{aligned}
y_u' &= y_0' - \frac{1}{18}h(-25f_u + 36f_v - 19f_w + 2f_1), \\
y_v' &= y_0' - \frac{1}{64}h(-93f_u + 120f_v - 66f_w + 7f_1), \\
y_w' &= y_0' - \frac{1}{9}h(-13f_u + 16f_v - 10f_w + f_1).
\end{aligned}
\tag{14}
$$

Taking a small step size $h$, and considering all the formulas in (9)–(11) for $n = 1, 2, \ldots, N - 1$, together with the ones developed in (12)–(14) for the first step, we obtain a global method that can provide accurate approximations to complete the integration along the interval $[0, x_N]$.

## 3. Characteristics of the Method

The main properties of the proposed technique PHNT are studied here, where the most challenging task is to analyze the convergence of the global method.

### 3.1. Consistency and Order of the Formulas

The formulas in (9)–(11) may be written as:

$$
\bar{A}\,V_n = h\,\bar{B}\,V_n' + h^2\,\bar{D}\,F_n,
\tag{15}
$$

where $\bar{A}, \bar{B}, \bar{D}$ are constant matrices containing the coefficients of the formulas (9)–(11), and:

$$
\begin{aligned}
V_n &= (y_n, y_{n+u}, y_{n+v}, y_{n+w}, y_{n+1})^T, \\
V_n' &= (y_n', y_{n+u}', y_{n+v}', y_{n+w}', y_{n+1}')^T, \\
F_n &= (f_n, f_{n+u}, f_{n+v}, f_{n+w}, f_{n+1})^T.
\end{aligned}
$$

Using standard strategies (see [21]), assuming that $y(x)$ has enough derivatives, we define the operator $\ell$ related to the formulas in (9)–(11):

$$
\ell[y(x); h] = \sum_{j \in I}\left[\alpha_j y(x_n + jh) - h\beta_j y'(x_n + jh) - h^2\gamma_j y''(x_n + jh)\right],
\tag{16}
$$

where $\alpha_j, \beta_j$, and $\gamma_j$ are, respectively, vector columns of $\bar{A}, \bar{B}$ and $\bar{D}$, while $I$ denotes the set of indices, $I = \{0, u, v, w, 1\}$. Expanding in Taylor series about $x_n$, we obtain:

$$
\ell[y(x); h] = C_0 y(x_n) + C_1 h y'(x_n) + C_2 h^2 y''(x_n) + \cdots + C_q h^q y^q(x_n) + \ldots,
\tag{17}
$$

where:

$$C_q = \frac{1}{q!}\left[\sum_{j\in I}^{k} j^q \alpha_j - q\sum_{j\in I}^{k} j^{q-1}\beta_j - q(q-1)\sum_{j\in I}^{k} j^{q-2}\gamma_j\right], \quad q = 0,1,2,3,\ldots. \quad (18)$$

According to [16], the above operator and the associated formulas are said to be of order $p$ if $C_0 = C_1 = \ldots = C_{p+1} = 0, C_{p+2} \neq 0$, with $C_{p+2}$ as the vector of local truncation errors. For the formulas in (9)–(11), we obtain $C_0 = C_1 = \cdots = C_6 = 0$ and:

$$C_7 = \left(\frac{1}{181,440}, -\frac{1}{1,088,640}, \frac{67}{44,089,920}, \frac{1}{362,880}, \frac{11}{2,755,620}, \frac{1}{131,220}, \frac{1}{138,240}, \frac{1}{131,220}\right)^T,$$

This shows that each of the above formulas is of order 5. Since the order of the formulas is greater than one, they are consistent. For the ad hoc formulas used for the first step, it is easy to see that they are also consistent.

### 3.2. Convergence Analysis

We start by defining convergence, then we will show that the proposed method is convergent by writing all the formulas in (9)–(14) in an appropriate matrix-vector form.

**Definition 1.** *Let $y(x)$ denote the exact solution of the given singular boundary value problem and let $\{y_j\}_{j=0}^{N}$ be the approximations obtained with the developed numerical strategy. The method is said to be convergent of order p if, for a sufficiently small h, there exists a constant C independent of h, such that:*

$$\max_{0\leq j\leq N} |y(x_j) - y_j| \leq Ch^p.$$

Note that in this situation, we obtain $\max\limits_{0\leq j\leq N} |y(x_j) - y_j| \to 0$ as $h \to 0$.

**Theorem 1** (Convergence theorem). *Let $y(x)$ be the true solution of the SBVP in (1) with the boundary conditions in (2), and $\{y_j\}_{j=0}^{N}$ the discrete solution provided by the proposed global method. Then, the proposed method is convergent to order six.*

**Proof.** Following [22], we define the matrix $D$ of dimension $8N \times 8N$ given by:

$$D = \begin{bmatrix} D_{1,1} & D_{1,2} & \ldots & D_{1,2N} \\ \vdots & \vdots & & \vdots \\ D_{2N,1} & D_{2N,2} & \ldots & D_{2N,2N} \end{bmatrix},$$

where the elements $D_{i,j}$ are $4 \times 4$ sub-matrices, except the $D_{i,N+1}, i = 1,\ldots,2N$, which have a size of $4 \times 3$, and $D_{i,2N}, i = 1,\ldots,2N$, which have a size of $4 \times 5$. Those sub-matrices are

$$D_{i,i} = I, i = N+2,\ldots,2N, \text{ being } I \text{ the identity matrix of order four,}$$

$$D_{N,N} = \begin{bmatrix} 1 & 0 & 0 \\ 0 & 1 & 0 \\ 0 & 0 & 1 \\ 0 & 0 & 0 \end{bmatrix}; \quad D_{i,i-1} = \begin{bmatrix} 0 & 0 & 0 & -1 \\ 0 & 0 & 0 & -1 \\ 0 & 0 & 0 & -1 \\ 0 & 0 & 0 & -1 \end{bmatrix}, i = N+2,\ldots,2N;$$

$$
D_{N+1,N+1} = \begin{bmatrix} -1 & 1 & 0 & 0 & 0 \\ -1 & 0 & 1 & 0 & 0 \\ -1 & 0 & 0 & 1 & 0 \\ -1 & 0 & 0 & 0 & 1 \end{bmatrix}; \qquad D_{1,N+1} = h \begin{bmatrix} -\dfrac{1}{3} & 0 & 0 & 0 & 0 \\ -\dfrac{1}{2} & 0 & 0 & 0 & 0 \\ -\dfrac{2}{3} & 0 & 0 & 0 & 0 \\ -1 & 0 & 0 & 0 & 0 \end{bmatrix};
$$

$$
D_{i,N+i} = h \begin{bmatrix} 0 & 0 & 0 & -\dfrac{1}{3} \\ 0 & 0 & 0 & -\dfrac{1}{2} \\ 0 & 0 & 0 & -\dfrac{2}{3} \\ 0 & 0 & 0 & -1 \end{bmatrix}, i = 1\ldots,N-1; \qquad D_{N,2N} = h \begin{bmatrix} 0 & 0 & 0 & 0 & -\dfrac{1}{3} \\ 0 & 0 & 0 & 0 & -\dfrac{1}{2} \\ 0 & 0 & 0 & 0 & -\dfrac{2}{3} \\ 0 & 0 & 0 & 0 & -1 \end{bmatrix}.
$$

For the rest of the submatrices not included above, it is $D_{i,j} = \mathbb{O}$—that is, they are null matrices.

We also define the matrix $U$ of dimension $8N \times (4N+1)$:

$$
U = \begin{bmatrix} U_{1,1} & U_{1,2} & \ldots & U_{1,N} \\ \vdots & \vdots & & \vdots \\ U_{2N,1} & U_{2N,2} & \ldots & U_{2N,N} \end{bmatrix},
$$

where the elements $U_{i,j}$ are $4 \times 4$ submatrices except the $U_{i,1}, i = 1,\ldots,2N$, which have a size of $4 \times 5$. These submatrices are given as follows:

$$
U_{1,1} = h \begin{bmatrix} 0 & -\dfrac{329}{1080} & \dfrac{194}{405} & -\dfrac{139}{540} & \dfrac{89}{3240} \\ 0 & -\dfrac{87}{160} & \dfrac{193}{240} & -\dfrac{69}{160} & \dfrac{11}{240} \\ 0 & -\dfrac{106}{135} & \dfrac{448}{405} & -\dfrac{82}{135} & \dfrac{26}{405} \\ 0 & -\dfrac{51}{40} & \dfrac{26}{15} & -\dfrac{21}{20} & \dfrac{11}{120} \end{bmatrix};
$$

$$
U_{i,i} = h \begin{bmatrix} -\dfrac{2}{27} & \dfrac{28}{405} & -\dfrac{29}{1080} & \dfrac{1}{540} \\ -\dfrac{207}{1280} & \dfrac{1}{8} & -\dfrac{63}{1280} & \dfrac{13}{3840} \\ -\dfrac{34}{135} & \dfrac{64}{405} & -\dfrac{2}{27} & \dfrac{2}{405} \\ -\dfrac{9}{20} & \dfrac{4}{15} & -\dfrac{9}{40} & 0 \end{bmatrix}, i = 2\ldots,N;
$$

$$
U_{i,i-1} = h \begin{bmatrix} 0 & 0 & 0 & -\dfrac{83}{3240} \\ 0 & 0 & 0 & -\dfrac{163}{3840} \\ 0 & 0 & 0 & -\dfrac{8}{135} \\ 0 & 0 & 0 & -\dfrac{11}{120} \end{bmatrix}, i = 3,\ldots,N; \qquad U_{2,1} = h \begin{bmatrix} 0 & 0 & 0 & 0 & -\dfrac{83}{3240} \\ 0 & 0 & 0 & 0 & -\dfrac{163}{3840} \\ 0 & 0 & 0 & 0 & -\dfrac{8}{135} \\ 0 & 0 & 0 & 0 & -\dfrac{11}{120} \end{bmatrix};
$$

$$U_{N+1,1} = \begin{bmatrix} 0 & -\frac{25}{18} & 2 & -\frac{19}{18} & \frac{1}{9} \\ 0 & -\frac{93}{64} & \frac{15}{8} & -\frac{33}{32} & \frac{7}{64} \\ 0 & -\frac{13}{9} & \frac{16}{9} & -\frac{10}{9} & \frac{1}{9} \\ 0 & -\frac{3}{2} & 2 & -\frac{3}{2} & 0 \end{bmatrix};$$

$$U_{N+j,j} = \begin{bmatrix} -\frac{19}{40} & \frac{152}{405} & -\frac{17}{120} & \frac{31}{3240} \\ -\frac{351}{640} & \frac{4}{15} & -\frac{81}{640} & \frac{17}{1920} \\ -\frac{8}{15} & \frac{64}{405} & -\frac{1}{5} & \frac{4}{405} \\ -\frac{27}{40} & \frac{8}{15} & -\frac{27}{40} & -\frac{11}{120} \end{bmatrix}, j = 2,\ldots,N;$$

$$U_{N+j,j-1} = \begin{bmatrix} 0 & 0 & 0 & -\frac{329}{3240} \\ 0 & 0 & 0 & -\frac{193}{1920} \\ 0 & 0 & 0 & -\frac{41}{405} \\ 0 & 0 & 0 & -\frac{11}{120} \end{bmatrix}, j = 3,\ldots,N; \qquad U_{N+2,1} = \begin{bmatrix} 0 & 0 & 0 & 0 & -\frac{329}{3240} \\ 0 & 0 & 0 & 0 & -\frac{193}{1920} \\ 0 & 0 & 0 & 0 & -\frac{41}{405} \\ 0 & 0 & 0 & 0 & -\frac{11}{120} \end{bmatrix}.$$

For the rest of submatrices $U_{i,j}$ not included above, it is $U_{i,j} = \mathbb{O}$—that is, they are null matrices.

We note that the submatrices $D_{i,j}$ and $U_{i,j}$ contain the coefficients of the formulas in (12)–(14) and those of the formulas in (9)–(11), for $n = 1, 2, \ldots, N-1$.

Let us denote the vectors of exact values as:

$$Y = \left(y(x_u), y(x_v), y(x_w), y(x_1), \ldots, y(x_{N-1+w}), y'(x_0), y'(x_u), \ldots, y'(x_N)\right)^T,$$

$$F = \left(f(x_0, y(x_0), y'(x_0)), f(x_u, y(x_u), y'(x_u)), \ldots, f(x_N, y(x_N), y'(x_N))\right).$$

Note that $Y$ has $(4N-1) + (4N+1) = 8N$ components while $F$ has $(4N+1)$ components, because, due to the boundary conditions in (2), $y(x_0)$ and $y(x_N)$ are known values, $y(x_0) = y_a, y(x_N) = y_b$.

The exact form of the discretized formulas to approximate the boundary value problem can be written as:

$$D_{8N\times 8N}Y_{8N} + hU_{8N\times(4N+1)}F_{4N+1} + C_{8N} = L(h)_{8N}, \tag{19}$$

where we have included the dimensions for clarity. Here, $C_{8N}$ is a vector that contains the known values—that is:

$$C_{8N} = (-y_a, -y_a, -y_a, -y_a, 0, \ldots, 0, y_b, 0, \ldots, 0)^T,$$

and $L(h)_{8N}$ is another vector containing the LTEs of the formulas, given by:

$$L(h)_{8N} \simeq \begin{bmatrix} \frac{83h^6 y^{(6)}(x_0)}{699,840} + \mathcal{O}(h^7) \\ \frac{163h^6 y^{(6)}(x_0)}{829,440} + \mathcal{O}(h^7) \\ \frac{h^6 y^{(6)}(x_0)}{3645} + \mathcal{O}(h^7) \\ \frac{11h^6 y^{(6)}(x_0)}{25,920} + \mathcal{O}(h^9) \\ \frac{67h^7 y^{(7)}(x_1)}{44,089,920} + \mathcal{O}(h^8) \\ \frac{h^7 y^{(7)}(x_1)}{362,880} + \mathcal{O}(h^9) \\ \frac{11h^7 y^{(7)}(x_1)}{2,755,620} + \mathcal{O}(h^8) \\ \frac{h^7 y^{(7)}(x_1)}{181,440} + \mathcal{O}(h^{10}) \\ \vdots \\ \frac{h^7 y^{(7)}(x_{N-1})}{181,440} + \mathcal{O}(h^8) \\ \frac{329h^5 y^{(6)}(x_0)}{699,840} + \mathcal{O}(h^7) \\ \frac{193h^5 y^{(6)}(x_0)}{414,720} + \mathcal{O}(h^7) \\ \frac{41h^5 y^{(6)}(x_0)}{87,480} + \mathcal{O}(h^7) \\ \frac{11h^5 y^{(6)}(x_0)}{25,920} + \mathcal{O}(h^7) \\ \frac{h^6 y^{(7)}(x_1)}{131,220} + \mathcal{O}(h^7) \\ \frac{h^6 y^{(7)}(x_1)}{138,240} + \mathcal{O}(h^7) \\ \frac{h^6 y^{(7)}(x_1)}{131,220} + \mathcal{O}(h^7) \\ -\frac{h^7 y^{(8)}(x_1)}{1,088,640} + \mathcal{O}(h^8) \\ \vdots \\ -\frac{h^{10} y^{(8)}(x_{N-1})}{1,088,640} + \mathcal{O}(h^{11}) \end{bmatrix}.$$

Concerning the approximate values, they are provided by the system:

$$D_{8N \times 8N} \bar{Y}_{8N} + hU_{8N \times (4N+1)} \bar{F}_{4N+1} + C_{8N} = 0, \tag{20}$$

where $\bar{Y}_{8N}$ approximates the vector $Y_{8N}$—that is:

$$\bar{Y}_{8N} = \left( y_u, y_v, y_w, y_1, \ldots, y_{N-1+w}, y_0', y_u', \ldots, y_N' \right)^T,$$

In addition:

$$\bar{F}_{4N+1} = \left( f_0, f_u, f_v, f_w, f_1, \ldots, f_N \right)^T.$$

We subtract (20) from (19) to obtain:

$$D_{8N \times 8N} E_{8N} + hU_{8N \times (4N+1)} (F - \bar{F})_{4N+1} = L(h)_{8N}, \tag{21}$$

where $E_{8N} = Y_{8N} - \bar{Y}_{8N} = \left( e_u, e_v, \ldots, e_{N-1+w}, e_0', e_u', \ldots, e_N' \right)^T$ is a vector of errors at the discrete points.

Through the Mean-Value Theorem in [23], we can put any convenient subindex $i$ as:

$$f(x_i, y(x_i), y'(x_i)) - f(x_i, y_i, y_i') = (y(x_i) - y_i) \frac{\partial f}{\partial y} (\xi_i) + (y'(x_i) - y_i') \frac{\partial f}{\partial y'} (\xi_i),$$

where $\xi_i$ denotes intermediate points in the line between $(x_i, y(x_i), y'(x_i))$ and $(x_i, y_i, y'_i)$. Thus, we can say that:

$$(F - \bar{F})_{4N+1} \; = \; \begin{pmatrix} \frac{\partial f}{\partial y}(\xi_0) & 0 & \cdots & 0 & \frac{\partial f}{\partial y'}(\xi_0) & 0 & \cdots & 0 \\ 0 & \frac{\partial f}{\partial y}(\xi_u) & \cdots & 0 & 0 & \frac{\partial f}{\partial y'}(\xi_u) & \cdots & 0 \\ \vdots & \vdots & \ddots & \vdots & \vdots & \vdots & \ddots & \vdots \\ 0 & 0 & \cdots & \frac{\partial f}{\partial y}(\xi_N) & 0 & 0 & \cdots & \frac{\partial f}{\partial y'}(\xi_N) \end{pmatrix} \begin{pmatrix} e_0 \\ e_u \\ \vdots \\ e_N \\ e'_0 \\ e'_u \\ \vdots \\ e'_N \end{pmatrix}$$

$$= \; \begin{pmatrix} 0 & \cdots & 0 & \frac{\partial f}{\partial y'}(\xi_0) & 0 & \cdots & 0 & 0 \\ \frac{\partial f}{\partial y}(\xi_u) & \cdots & 0 & 0 & \frac{\partial f}{\partial y'}(\xi_u) & \cdots & 0 & 0 \\ \vdots & \ddots & 0 & \vdots & \vdots & \ddots & \vdots & \vdots \\ 0 & \cdots & \frac{\partial f}{\partial y}(\xi_{N-1+w}) & 0 & 0 & \cdots & \frac{\partial f}{\partial y'}(\xi_{N-1+w}) & 0 \\ 0 & \cdots & 0 & 0 & 0 & \cdots & 0 & \frac{\partial f}{\partial y'}(\xi_N) \end{pmatrix} \begin{pmatrix} e_u \\ e_v \\ \vdots \\ e_{N-1+w} \\ e'_0 \\ e'_u \\ \vdots \\ e'_N \end{pmatrix}$$

$$= \; J_{(4N+1) \times 8N} \, E_{8N} .$$

Note that in the second identity we have used, $e_0 = y(x_0) - y_a = 0$ and $e_N = y(x_N) - y_b = 0$.

In view of the above, the equation in (21) may be arranged as:

$$\left( D_{8N \times 8N} + h U_{8N \times (4N+1)} J_{(4N+1) \times 8N} \right) E_{8N} = L(h)_{8N}, \tag{22}$$

Setting $M = D + hUJ$, we simply find that:

$$M_{8N \times 8N} E_{8N} = L(h)_{8N}. \tag{23}$$

Following [24], we prove that, except for a few selected values of $h > 0$, matrix $M$ is invertible. If we use the abbreviated notation $D_N = D_{8N \times 8N}$, given the form of this matrix where the submatrices have many zeros, it is easy to verify that, for $N = 2$, the determinant is $|D_2| = -2h$. Now, by induction, it can be proven that $|D_N| = -Nh$; thus $D_N$ is invertible as long as it is $h > 0$.

Now, the matrix $M$ may be rewritten as:

$$M = D + hUJ = (Id - B)D,$$

where $Id$ is the identity matrix of order $8N$ and $B = -hUJD^{-1}$. Thus, we find that $|M| = |Id - B| \, |D|$.

As $|\lambda Id - B| = \prod_{i=1}^{8N} (\lambda - \lambda_i)$ is the characteristic polynomial of $B$, in order to have $|Id - B| \neq 0$, if we take $\lambda = 1$ it is sufficient to choose $h$, such that:

$$\left\{ 1/\bar{\lambda}_i : \bar{\lambda}_i \text{ is an eigenvalue of } UJD^{-1} \right\}.$$

For such values of $h$, the equation in (23) may be rewritten as:

$$E = \left( M^{-1} \right) L(h). \tag{24}$$

The maximum norm in $\mathbb{R}$, $\|E\| = \max_i |e_i|$, and the corresponding matrix-induced norm in $\mathbb{R}^{8N \times 8N}$ are considered. If we expand the terms of $M^{-1}$ in powers of $h$, it can be shown that $\|M^{-1}\| = \mathcal{O}(h^{-2})$.

Assuming that $y(x)$ has in $[0, x_N]$ bounded derivatives up to the necessary order, from (24) and the vector $L(h)$ of local truncation errors, we can obtain:

$$
\begin{aligned}
\|E\| &\leq \|\left(M^{-1}\right)\| \, \|L(h)\| \\
&= \mathcal{O}(h^{-2})\, \mathcal{O}(h^7) \\
&\leq K h^5.
\end{aligned}
$$

We have shown that the global method exhibits a fifth-order convergence. Nevertheless, in view of the form of the vector $L(h)$, we see that, assuming the sufficient smoothness of the solution, at the mesh points we obtain a superconvergence order (see Ascher et al. [25]):

$$
|e_j| = |y(x_j) - y_j| \leq |\mathcal{O}(h^{-2})|\,|\mathcal{O}(h^8)| \leq K h^6, \quad j = 1, 2, \ldots N.
$$

Therefore, the proposed method is convergent, providing sixth-order approximations. $\quad\square$

## 4. Implementation Issues

The PHNT is implemented in a block unification mode. We rewrite the systems in (20) as $F(y) = 0$ and the unknowns as:

$$
\tilde{\mathbf{U}} = \bigcup \{y_j\}_{j=1,\ldots,N-1} \bigcup \{y'_j\}_{j=0,\ldots,N} \bigcup \{y_{j+u}, y_{j+v}, y_{j+w}, y'_{j+u}, y'_{j+v}, y'_{j+w}\}_{j=0,1,\ldots,N-1}.
$$

Then, we use Modified Newton's method (MNM) to solve non-linear equations, since the PHNT is an implicit scheme. The MNM is given by:

$$
\tilde{\mathbf{U}}^{i+1} = \tilde{\mathbf{U}}^i - \left(\mathbf{J}^i\right)^{-1}\mathbf{F}^i,
$$

where $\mathbf{J}$ represents the jacobian matrix of $\mathbf{F}$. The starting values for using MNM for solving the systems given in (12)–(14) for each iteration are taken as those provided by the linear interpolation obtained throughout the boundary values, while the stopping criterion considers a maximum number of 100 iterations and an error between two successive approximations of less than $10^{-16}$.

We enumerate and summarize how the PHNT is utilized to give numerical solutions to physical models and catalytic diffusion–reaction problems as follows:

1.  Let us take $N > 0 \in \mathbb{N}$, and define $h = \frac{x_N - x_0}{N}$ to generate the partition:

$$
P_N = \bigcup \{x_j\}_{j=0,1,\ldots,N} \bigcup \{x_{j+k}\}_{k=u,v,w;\,j=0,1,\ldots,N-1}.
$$

2.  Using Equations (12)–(14) and Equations (9)–(11) for $n = 1, \ldots, N-1$, we can form a system of equations with variables:

$$
\{y_u, y_v, y_w, y'_u, y'_v, y'_w\} \bigcup \{y_j\}_{j=1,\ldots,N-1} \bigcup \{y'_j\}_{j=0,\ldots,N} \bigcup \{y_{j+u}, y_{j+v}, y_{j+w}, y'_{j+u}, y'_{j+v}, y'_{j+w}\}_{j=1,\ldots,N-1}.
$$

3.  We make just one block matrix equation by joining all the equations generated in the previous step of the partition $P_N$ with the given boundary conditions.
4.  We solve the single block matrix equation simultaneously to obtain the approximate solutions for the SBVP on the whole interval $[x_0, x_N]$.

## 5. Numerical Illustrations

This section presents the numerical outcomes and discussion of the proposed PHNT for the solution of the singular physical models and catalytic diffusion–reaction problems of the form (1). The accuracy of the PHNT is measured by utilizing the following formulas:

$$
ABER = \|y(x_j) - y_j\|, \qquad MAXABER = \max_{j=0,1,\ldots,N} \|y(x_j) - y_j\|,
$$

where ABER denotes the absolute error at the considered node, MAXABER is the maximum absolute error along the considered interval, $y(x_j)$ is the theoretical solution, and $y_j$ is the approximate solution provided by the PHNT.

### 5.1. Example 1

We firstly consider the following scalar Lane–Emden singular equation (SCLSE), which corresponds to the reaction–diffusion process in a spherical permeable catalyst as reported in [3,26],

$$y''(x) + \frac{2}{x}y'(x) - \phi^2 y(x)^n = 0, \quad y(1) = 1 \text{ (at the catalyst surface)}, \tag{25}$$
$$y'(0) = 0 \text{ (at the centre of the catalyst surface)}, \quad x \in [0,1].$$

The general analytical solution of problem (25) is unknown, but its solution for $n = 1$, is given by $y(x) = \frac{\sinh(x\phi)}{x\sinh(\phi)}$, where $\phi$ is the Thiele modulus and $\phi^2 = \frac{\text{reaction rate at the catalyst surface}}{\text{diffusion rate at the catalyst pores}}$.

Table 1 presents the numerical results with the proposed method, showing that they are very close to the theoretical solution available for $n = 1$. The CPU time with the PHNT for the value of $N = 10$ in Table 1 is 0.2188 s.

**Table 1.** Comparison of *PHNT* and the exact solution on test 1 with $h = \frac{1}{10}, n = 1, \phi = 5$.

| $x$ | *PHNT* | **Exact Solution** | **ABER** |
|---|---|---|---|
| 0.1 | 0.07022543735304641 | 0.07022543922779090 | $1.87474 \times 10^{-9}$ |
| 0.2 | 0.07918802960817403 | 0.07918802869127974 | $9.16894 \times 10^{-10}$ |
| 0.3 | 0.09565082522028227 | 0.09565082330512954 | $1.91515 \times 10^{-9}$ |
| 0.4 | 0.12219351619982496 | 0.12219351358270766 | $2.61712 \times 10^{-9}$ |
| 0.5 | 0.16307123522003290 | 0.16307123192997786 | $3.29006 \times 10^{-9}$ |
| 0.6 | 0.22500992040856140 | 0.22500991644891966 | $3.95964 \times 10^{-9}$ |
| 0.7 | 0.31848116606908733 | 0.31848116156439320 | $4.50469 \times 10^{-9}$ |
| 0.8 | 0.45971591487821106 | 0.45971591027923203 | $4.59898 \times 10^{-9}$ |
| 0.9 | 0.67387038375554700 | 0.67387038020431460 | $3.55123 \times 10^{-9}$ |
| 1 | 1.00000000000000000 | 1.0000000000000000 | 0.00000 |

To analyze the impact of the Thiele modulus $(\phi)$ on the concentration profile $(y(x))$, we also considered other values of $\phi$ and $n$. Figure 1 displays the numerical outcomes for various values of $\phi$ and $n$. We observed that in Figure 1, the concentration profile increases when $\phi$ diminishes.

### 5.2. Example 2

As a second test problem, we consider the non-homogeneous SCLSE, which corresponds to the physical model problem 2 in [4]:

$$y''(x) + \frac{1}{x}y'(x) + y(x) = 4 - 9x + x^2 - x^3, \quad y(0) = 0, y(1) = 0 \tag{26}$$

where the exact solution is given by $y(x) = x^2 - x^3$.

We have applied PHNT to test problem 2; Figure 2 presents the numerical solution of PHNT, which is very close to the exact solution for problem 2. Figure 3 shows a graphical representation of the absolute errors (ABER) for different values of $x$. The CPU time with the PHNT for the value of $N = 10$ in Figures 2 and 3 is 0.0625 s.

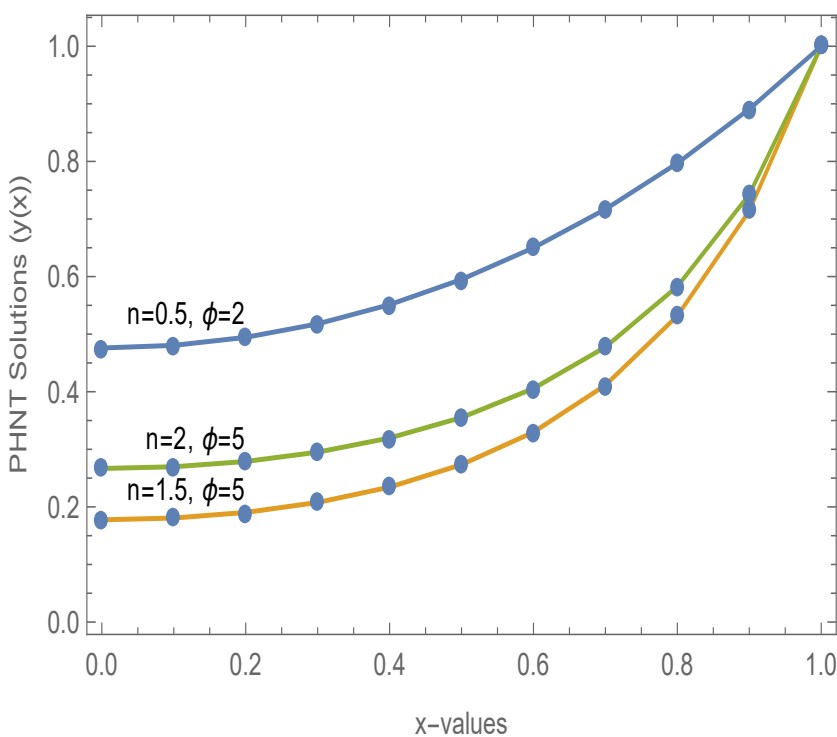

**Figure 1.** Approximate solutions of PHNT for $h = \frac{1}{10}$ for different values of $\phi$ and $n$ for Example 1.

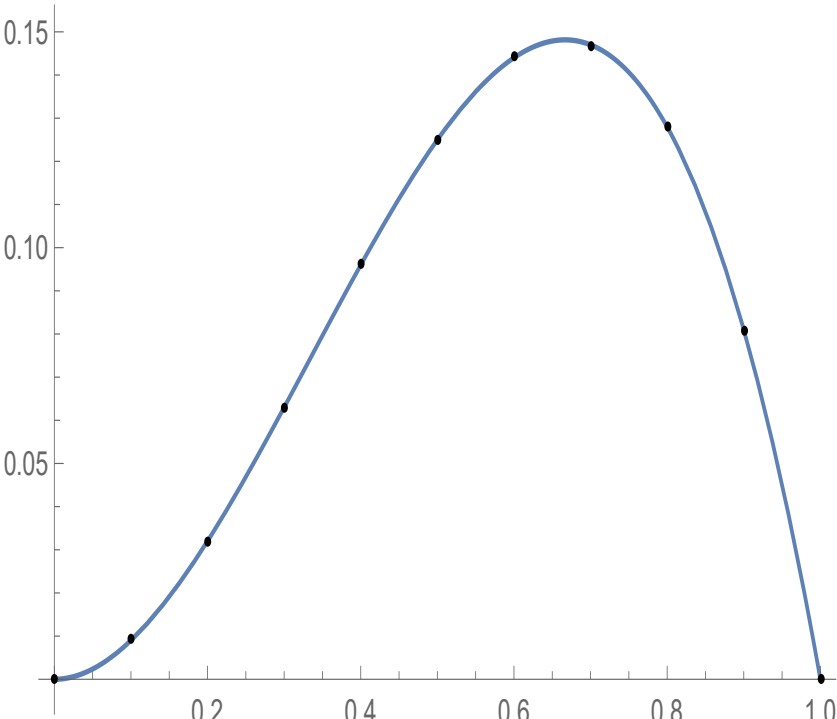

**Figure 2.** Exact solution and the discrete one obtained with the PHNT for $h = \frac{1}{10}$ for Example 2.

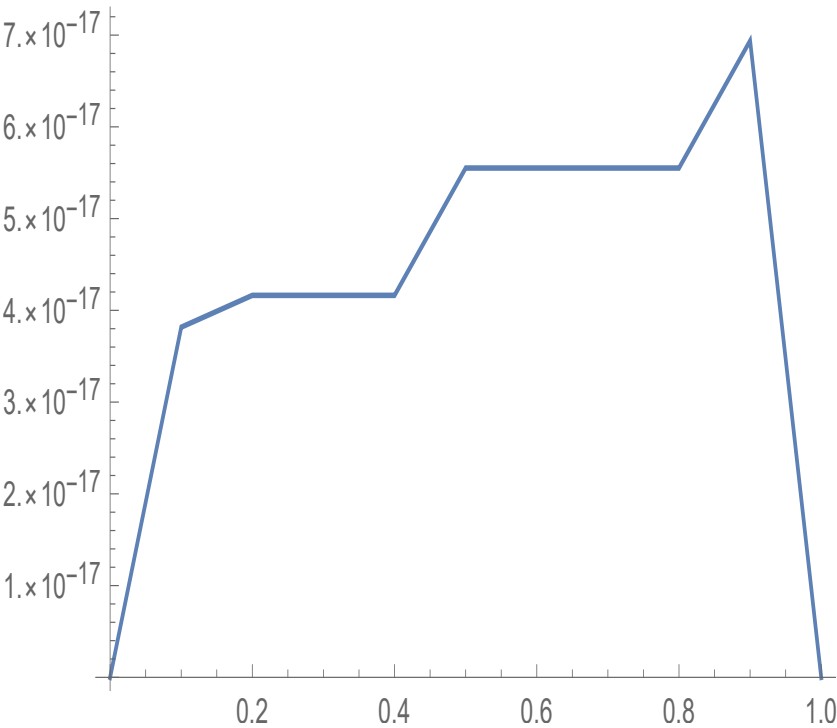

**Figure 3.** Plot of the ABER with PHNT for $h = \frac{1}{10}$ for Example 2.

### 5.3. Example 3

Let us consider the non-linear SCLSE, which corresponds to the physical model problem of thermal explosion in cylindrical vessel reported by Thula and Roul [3] and Roul et al. [14]:

$$y''(x) + \frac{1}{x} y'(x) = \exp(y(x)), \quad y(0) = 0, y'(1) = 0, \tag{27}$$

where the analytical solution is given by $y(x) = 2 \log \left( \frac{2\sqrt{6}+1-5}{(2\sqrt{6}-5)x^2+1} \right)$.

Test problem 3 is numerically solved using the new *PHNT* for different values of step size $(h)$. The numerical results and the comparisons of MAXABER and ABER between *PHNT* and the methods in [3,14] are abridged in Table 2.

We note that the *PHNT* method presents a better performance compared with the techniques in [3,14]. Additionally, the CPU time in our proposed *PHNT* to obtain the approximate solutions for problem 3 with step size $h = \frac{1}{16}$ is 0.4531 s. In Table 2, we have included the numerical rate of convergence (ROC) with the following formula:

$$ROC \simeq -\log_2 \left( \frac{MAXABER_h}{MAXABER_{2h}} \right).$$

### 5.4. Example 4

As a test problem 4, we consider a non-linear homogeneous SCLSE, which corresponds to the physical model problem arising in chemistry and chemical kinetics. The formulation of heat and mass transfer within porous catalyst particles is reported by Ravikanth [27]:

$$y''(x) + \frac{\lambda}{x} y'(x) - \phi^2 y(x) \exp \left( \frac{rs(1-y(x))}{c(1-y(x))+1} \right) = 0, \quad y(0) = 0, y'(1) = 0, \tag{28}$$

where the analytical solution is unknown.

**Table 2.** Maximum absolute errors (MAXABER) for test problem 3.

| $h$ | Method | $MAXABER$ | $ROC$ |
|---|---|---|---|
| $\frac{1}{8}$ | *PHNT* | $1.91969 \times 10^{-10}$ | |
| $\frac{1}{8}$ | Method in [3] | $1.01400 \times 10^{-8}$ | |
| $\frac{1}{8}$ | Method in [14] | $8.53810 \times 10^{-10}$ | |
| $\frac{1}{16}$ | *PHNT* | $2.99397 \times 10^{-12}$ | 6.00 |
| $\frac{1}{16}$ | Method in [3] | $5.80500 \times 10^{-10}$ | 4.13 |
| $\frac{1}{16}$ | Method in [14] | $2.19100 \times 10^{-11}$ | 5.30 |
| $\frac{1}{32}$ | *PHNT* | $4.77118 \times 10^{-14}$ | 5.97 |
| $\frac{1}{32}$ | Method in [3] | $3.49500 \times 10^{-11}$ | 4.05 |
| $\frac{1}{32}$ | Method in [14] | $3.92400 \times 10^{-13}$ | 5.94 |

We solved problem 4 with the PHNT scheme for $\phi = r = s = c = 1$. The PHNT approximate solutions for $\lambda = 2$ and $\lambda = 4$ for different values of $x$ are plotted in Figure 4.

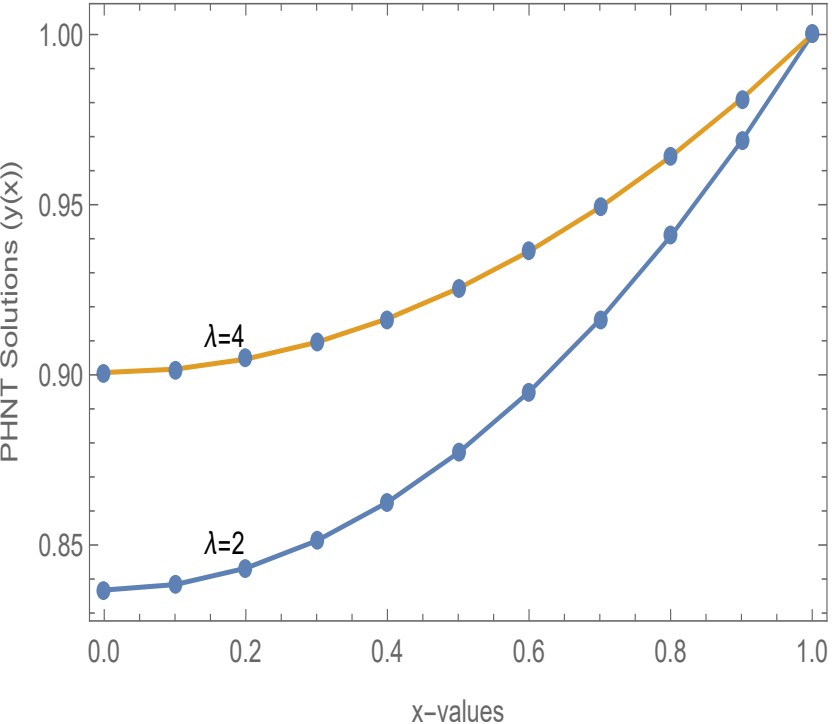

**Figure 4.** Approximate solutions of PHNT for $h = \frac{1}{10}$ for different values of $\lambda = 2$ and $\lambda = 4$ for test problem 4.

Table 3 contains the numerical results provided by PHNT. We also used, for comparison, the cubic spline method (CSM) and the modified Adomian decomposition technique (MADT) in [13,27], with the PHNT being significantly better. The CPU time used by PHNT for $N = 10$ with the specifications in Table 3 is 0.5938 s.

**Table 3.** Comparison of approximation solutions for test problem 4 for $\lambda = 2$.

| $x$ | $PHNT, h = \frac{1}{10}$ | Method in [13], $h = \frac{1}{10}$ | Method in [27], $h = \frac{1}{50}$ |
|---|---|---|---|
| 0.1 | 0.8383648968983356 | 0.838364878696000 | 0.83836491959750 |
| 0.2 | 0.8431842515107244 | 0.843184233589000 | 0.84318428772800 |
| 0.3 | 0.8512302074850839 | 0.851230190133000 | 0.85123026453741 |
| 0.4 | 0.8625224114697174 | 0.862522394979000 | 0.86252249405263 |
| 0.5 | 0.8770865272403503 | 0.877086511985000 | 0.87708663616863 |
| 0.6 | 0.8949523006678164 | 0.894952287104000 | 0.89495243141739 |
| 0.7 | 0.9161509382771176 | 0.916150926969000 | 0.91615107927542 |
| 0.8 | 0.9407117001197036 | 0.940711691749000 | 0.94071183074544 |
| 0.9 | 0.9686575885218047 | 0.968657583887000 | 0.96865767679753 |
| 1.0 | 1.0000000000000000 | 1.0000000000000000 | 1.000000000000000 |

## 6. Conclusions

This work presented a reliable PHNT approach for solving the scalar and system of SBVP of Lane-Emden type in various physical models and chemical kinetics. We employed a set of optimized hybrid block formulas given in (9)–(11) which are combined with an appropriate starting algorithm in (12)–(14) specifically designed to cope with the singularity at the beginning of the integration interval of the considered problem. Four real-world model problems in applied sciences and engineering are solved numerically to show the strength and quality of the proposed PHNT. The obtained approximate results in Tables 1–3 and Figures 1–4 accentuate the effectiveness of the new methodology. An interesting question to be addressed in future investigations is how to select the optimal stepsizes. Notice that we have used the same step size for both groups of formulas, but we could have chosen one stepsize for the formulas in (9)–(11) and another for the formulas in (12)–(14). How to obtain the optimal values for these step sizes is an open question.

**Author Contributions:** Conceptualization, M.A.R. and H.R.; methodology, M.A.R. and H.R.; validation, M.A.R. and H.R.; formal analysis, M.A.R. and H.R.; investigation, M.A.R. and H.R.; data curation, H.R.; writing—original draft preparation, M.A.R.; writing—review and editing, M.A.R. and H.R. All authors have read and agreed to the published version of the manuscript.

**Funding:** This research did not receive any funding.

**Institutional Review Board Statement:** Not applicable.

**Informed Consent Statement:** Not applicable.

**Data Availability Statement:** Not applicable.

**Conflicts of Interest:** The authors declare no conflict of interest.

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
