# Peer review of "Numerical Solution for Singular Boundary Value Problems Using a Pair of Hybrid Nyström Techniques"

_axioms, doi:10.3390/axioms10030202_

Round 1
Reviewer 1 Report
The use of omega and w together is confusing at some places.
Reviewer 2 Report
This manuscript presents an efficient pair of hybrid Nyström techniques to solve second order Lane-Emden singular boundary value problems directly.
Reviewer's comments:
1. The originality and the significance of the content of this paper are of average level.
2. I appreciated the quality of the presentation as being of a lower than average level, because the results should have been presented more succinctly and synthetically.
3. The scientific soundness is of average level and is ensured by the presented results.
4. I consider that the interest for readers is of average level.
5. In conclusion, the overall merit of this paper can be appreciated as "average".
I recommend this paper to be publish.
Reviewer 3 Report
In this paper the authors have applied the hybrid Nyström techniques to solve second order Lane-Emden singular boundary value problems directly. They have proved the convergence theorem. Solving some examples and comparing the results they have tried to show the abilities of the method. I have the following comments on this paper:
1- In the introduction, please add about the advantages of your method. In Line 34, you have described about it. But you need more clarification.
2- One of the main problems in this method is finding the optimal value of h. How do you want to do it? Recently I have studied some papers from Prof. Fariborzi Araghi related to the CESTAC method to find the optimal h in these methods. Please add some descriptions about it and cite some references.
3- Check all punctuations at the end of equations. In some cases they are wrong.
4- In your examples, you have the exact solution. In real life problems we do not have it. So how do you want to show the accuracy, when you do not have the exact solution?
5- In all examples, h is 1/10. Why?
6- Conclusion can be improved. Please add some future works.
7- Please highlight all corrections.
I propose a major revision.
One of the proposed strategies
3 uses three off-step points. The obtained formulas are paired with an appropriate set of formulas
4 implemented for the first step to avoid the singularity at the left end of the integration interval.
5 The fundamental properties of the proposed scheme are analyzed. Some test problems, including
6 chemical kinetics and physical model problems, are solved numerically to determine the efficiency
7 and validity of the proposed approach.
Round 2
Reviewer 3 Report
I do not have more comments on this paper.